# A bioinspired microdevice unifying energy storage and actuation through hydration control

Wenlan Zhang [1,2], Leandro Merces [1,2,5], Jiachen Ma [1,2], Christian Becker[1,2], Daniil Karnaushenko [1,2], Hongmei Tang[1,2], Jiang Qu [3], Letícia Mariê Minatogau Ferro [1,2], Yang Huang[4], Yaping Yan[1,2], Yeji Lee[1,2], Vineeth K. Bandari[1,2], Dmitriy D. Karnaushenko[1,2], Aleksandr I. Egunov [1,2], Minshen Zhu [1,2] ✉ & Oliver G. Schmidt [1,2] ✉

Biological systems seamlessly integrate energy storage and actuation within compact architectures, whereas synthetic approaches largely implement these functions as separate components. Conjugated polymers can couple both, yet their operation relies on ion insertion accompanied by hydration water within the polymer backbone, creating an intrinsic trade-off between performance and stability. Here we show that anion hydration governs this trade-off. In-operando Raman spectroscopy and time-resolved mass measurements reveal that reducing anion hydration suppresses water ingress, mitigates backbone degradation and converts the polymer response from a two-step swelling process into a single, rapid volumetric relaxation. Leveraging this principle, we realize a sub-millimetre monolithic device that integrates energy storage and actuation within a 0.56 mm$^2$ footprint. A centrally configured dual-cell microbattery delivers 161 mAh cm$^{-2}$ and reduces the energy consumption of surrounding actuators by fourfold. Hydration control, as the governing design parameter for multifunctional devices, holds translational promise for integrated energy–motion architectures at the microscale.

Biological systems inherently embody multifunctionality, integrating structural support, energy storage, and mechanical actuation to facilitate adaptation and resilience[1,2]. For instance, biological muscles localize both energy supply and mechanical output within the same structure (Fig. 1a)[3,4]. This design avoids delays associated with distant power transmission and reduces the risk of failure from disconnected components. Among artificial materials, conjugated polymers offer a promising platform for replicating such complexity within a single device[4,5]. The doping and extraction of ions into and from the polymer backbone simultaneously induce charge storage/

release and volumetric change (Fig. 1b). As a result, conjugated polymers inherently couple energy storage with actuation, making them suited for the development of integrated systems that mirror the multifunctional sophistication of biological materials[6,7].

To ensure stable ionic and electronic transport during electrochemical cycling, any microstructural changes in the polymer that occur during ion insertion and extraction must be fully reversible. Many efforts have been focused on investigating the ion impact on conjugated polymer structures[8–10]. However, ions are accompanied by bulky solvation shells, particularly in aqueous electrolytes, where a

[1]Material Systems for Nanoelectronics, Chemnitz University of Technology, Chemnitz, Germany. [2]Research Center for Materials, Architectures and Integration of Nanomembranes (MAIN), Chemnitz University of Technology, Chemnitz, Germany. [3]Leibniz Institute for Solid State and Materials Research (IFW Dresden), Dresden, Germany. [4]Advanced Materials Thrust, Function Hub, The Hong Kong University of Science and Technology (Guangzhou), Guangzhou, China. [5]Present address: Ilum School of Science, Brazilian Center for Research in Energy and Materials (CNPEM), Campinas-SP, Brazil. ✉e-mail: minshen.zhu@main.tu-chemnitz.de; oliver.schmidt@main.tu-chemnitz.de

single ion may carry a hydration shell containing up to six water molecules[11–13]. Strong swelling induced by water molecules facilitates electrochemical actuation, recognized as a low-power actuation technology with potential in microscale applications, such as micro-robots with limited power sources[14].

Despite its simple composition, water plays a vital role in influencing the behavior of conjugated polymers on both molecular and interfacial levels. It facilitates structural dynamics, alters physicochemical properties, and affects long-term stability[15]. Additionally, water reduces intermolecular interactions, hinders polymer chain connectivity, and affects charge transport, thereby limiting access to redox-active sites[16]. In severe cases, water absorption can lead to delamination from current collectors or even the dissolution of the charged polymer[17]. Nonetheless, the influence of water on the behavior of conjugated polymers is frequently overlooked or undervalued.

To elucidate the impact of water molecules on the stability of a representative conjugated polymer, polyaniline (PANI), we monitored water interactions within PANI in operando. Strong water interactions notably occur with the transition of PANI from electrochemically active forms to inert forms. Strong hydration forces water molecules into the PANI backbone, accelerating hydrolytic degradation and eventual failure[18,19]. We opted for triflate anions, known for disrupting hydration, to reduce water uptake and effectively suppress the hydrolytic degradation of PANI. The quantification of water molecules in PANI over the cycling process shows that instead of drawing water molecules into the PANI backbone, triflate anions repel water molecules away from the PANI backbone. Disrupted hydration and water withdrawal from the PANI backbone enable a monolithically stacked dual-cell microbattery with a footprint area of only 0.56 mm$^2$ to cycle stably for over 2200 times, achieving an unprecedented lifetime capacity of 161 mAh cm$^{-2}$, thus setting a benchmark for sub-millimeter batteries. Additionally, disrupting hydration smooths water-exchange dynamics, eliminating structural delays caused by rapid water intercalation under strong hydration, thereby promoting actuator relaxation and approximately 4-fold reduction in power consumption. Improved stability for electrochemical energy storage and enhanced efficiency of electrochemical actuation enable the design of a monolithically integrated sub-millimeter device that combines both functions. Additionally, the integrated device can serve as a power source for various electrical functions, including LED lighting and powering low-power digital watches.

## Results

Building on the model of biological muscles, we exploit electrochemical redox reactions in conjugated polymers to merge energy storage and actuation within a sub-millimeter device. Measuring below 1 mm$^2$, the sub-millimeter device (Fig. 1c) consists of a 0.56 mm$^2$ battery used for energy storage (Fig. 1d) and four legs connected to the microbattery (inset in Fig. 1c, detailed structure is shown in Supplementary Fig. 1) that act as micro-actuators. Achieving this high degree of integration poses a core challenge: at the sub-millimeter scale, polymer films must withstand repeated electrochemical and mechanical cycling, which demands careful thickness control for consistent performance. While thinner films afford greater flexibility and efficient actuation, their smaller mass significantly limits the charge storage ability of the microbattery. One solution is to stack multiple thin films to raise the load-to-area ratio[20]. However, this approach intensifies the need for precise layering of battery components, including metals for current collectors, anode materials, and spaces for electrolytes, all on a sub-millimeter footprint.

To address this challenge, we utilize a self-assembly method known as micro-origami[3,21–23], which employs standard photolithography to create thin films that fold into three-dimensional forms (Supplementary Fig. 2). The micro-origami process supports parallel fabrication, allowing 120 such stacks to be produced on a single

substrate (Fig. 1e). Figure 1f shows the precursor layout, consisting of two planar cathode–anode pairs for two sub-microbatteries. Regarding the microbattery, each cathode–anode pair comprises thin titanium (Ti, 10 nm) and gold (Au, 50 nm) films deposited on polyimide (PI) and sacrificial layer (SL) (top panel, Fig. 1f.1), fortified by SU-8 frames that protect the planar electrodes from warping during folding. The folding hinges on the SL are made of a hydrogel (HG) layer and a PI reinforcement. Once the SL is etched away, the HG swells in an alkaline solution (0.1 M NaOH), triggering the folding action. The planar precursor structure first becomes freestanding, and the folding of the central hinge is initiated. The inner two electrodes self-fold into a back-to-back contact. Subsequently, both side hinges are activated, ultimately folding the structure into two enclosed boxes that serve as individual battery units (the self-folding process is shown in Supplementary Movie 1). Notably, immersion in 0.1 M NaOH does not compromise the integrity of the PANI backbone, yet it still triggers a re-equilibration process when PANI functions as an electrode in the electrolyte (Supplementary Note 1, Supplementary Figs. 3 and 4). Moreover, the micro-origami method can integrate up to ten planar films into a five-layer stack (Supplementary Fig. 5), revealing its capacity for complex, high-order assemblies. Balancing process intricacy with the need for sufficient energy storage, we chose to implement a two-layer stack to form our microbattery (Supplementary Note 2, Supplementary Figs. 6 and 7).

Polyaniline (PANI) was deposited on the Ti/Au electrode as the cathode while zinc (Zn) was used as the anode. Layered Zn hexagonal sheets were formed after deposition (Supplementary Fig. 8), exposing a favorable (002) face for Zn reversibility[24,25]. Polymerization in an acidic medium yielded dark green PANI (emeraldine salt) with a porous nanofibrous morphology (Supplementary Fig. 8), offering abundant active sites and facilitating rapid charge transport. As shown in Fig. 2a, the water-rich solvation in ZnSO$_4$ electrolyte leads to rapid capacity degradation within just 20 cycles, whereas the Zn anode itself remains stable for more than 700 cycles (Supplementary Note 3, Supplementary Fig. 9). The galvanostatic charge–discharge plots show the disappearance of the discharging plateau (Supplementary Fig. 10), indicating the full degradation of PANI. The normalized cyclic voltammetry contour plots (Fig. 2b) show an irregular shift in redox peaks, suggesting instability in the redox processes. Notably, the oxidation peak near 0.4 V disappears, while a near-linear current rise emerges around 0.6 V, which is attributed to the onset of water decomposition[26]. On the cathodic side, an abnormal current increase is observed near -0.4 V, further indicating parasitic water reduction reactions[26]. However, the electrolyte itself is electrochemically stable within the voltage window used for the CV tests, as confirmed by the linear sweep voltammetry results shown in Supplementary Fig. 11. Nevertheless both Zn$^{2+}$ and SO$_4^{2-}$ ions exhibit strong hydration in aqueous environments[27,28], and density function theory (DFT) calculations also predict that SO$_4^{2-}$ will attract water molecules towards the PANI backbone (Supplementary Fig. 12). Therefore, the fast performance degradation implies that water-rich solvation shells play a central role. To test this hypothesis, we conducted in-operando Raman spectroscopy on PANI in a Zn||PANI full cell. In ZnSO$_4$ electrolyte, as the voltage exceeded 1.3 V (corresponding to approximately 0.3 V vs. Ag/AgCl), intense –OH stretching bands appeared in the 3000–3500 cm$^{-1}$ range (Fig. 2c), confirming extensive water participation. Simultaneously, the vibrational modes of the polymer backbone in the range of 1100 to 1800 cm$^{-1}$ show a drastic backbone rearrangement. The electrochemical reversibility of PANI relies on the stability of its conjugated backbone and the integrity of the redox-active emeraldine salt form, which has alternating benzenoid–quinoid units supporting charge delocalization and proton-coupled electron transfer[29]. Protonation of imine nitrogen allows for doping and charge transport, while structural degradation that is often initiated by imine hydrolysis disrupts π-

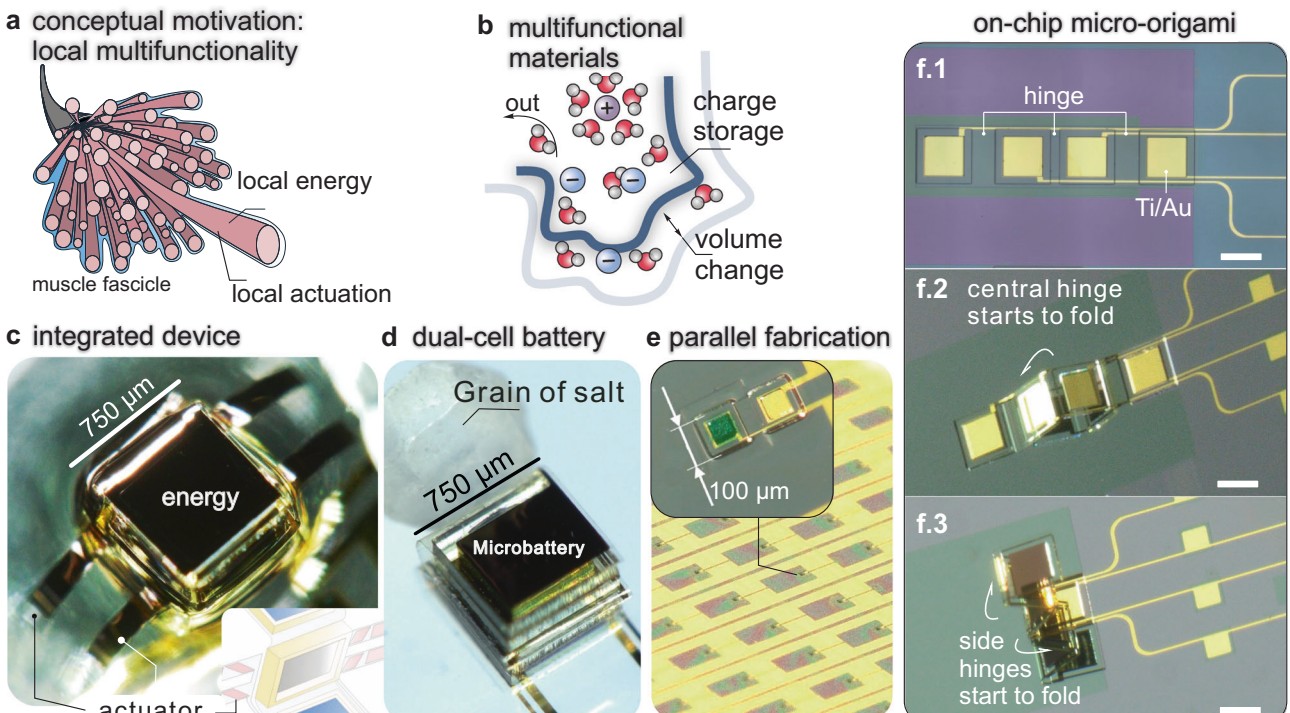

**Fig. 1 | Bio-inspired integration of energy storage and actuation. a** Schematic illustration of a biological muscle fascicle that monolithically integrates local energy and actuation. **b** Schematic illustration of ion doping/extraction in conjugated polymers, simultaneously enabling charge storage and volumetric change. **c** Optical image of an artificial sub-millimeter device integrating energy storage and actuation. The inset shows the integration layer, which consists of the actuators and the upper electrode in the bottom cell. **d** Optical image of the dual-cell microbattery with a footprint of 0.56 mm². **e** Optical image of parallel fabrication of 120 microstacks on a single substrate (22 × 22 mm²). **f** Optical images showing the micro-origami self-folding process of the dual-cell microbattery through a consecutive two-step folding. Scale bars: 500 μm. Panel **a** adapted from Servier Medical Art (https://smart.servier.com), licensed under CC BY 4.0 (https://creativecommons.org/licenses/by/4.0/).

conjugation and causes capacity loss[30]. The emergence of new peaks at 1490 cm⁻¹ (C=N stretching deformations) and 1591 cm⁻¹ (C=C stretching deformations of quinoid ring) reveals the formation of quinoid-imine structures, which are susceptible to hydrolysis and nucleophilic attack in aqueous environments, leading to the formation of p-benzoquinone and hydroquinone products due to water-induced degradation of PANI[31–33]. This is further supported by the increased proportion of C=O/C–O shown in X-ray photoelectron spectroscopy (XPS) results (Supplementary Note 4, Supplementary Fig. 13). The formation of *p*-benzoquinone and hydroquinone reflects irreversible structural decomposition of PANI, disrupting the conjugated backbone essential for charge transport, thereby leading to a sharp decline in electrochemical performance. Consistently, XPS analysis shows no evident change in $Zn^{2+}$ and $SO_4^{2-}$ signals after oxidation and reduction of the failed PANI sample, indicating that the polymer has become electrochemically inert. (Supplementary Note 5, Supplementary Fig. 14).

By replacing $SO_4^{2-}$ anions with triflate (OTf⁻), PANI shows a significant improvement in the cycling stability of over 100 cycles (Fig. 2a). Charge and discharge plateaus remain stable, and the polarization does not increase evidently (Supplementary Fig. 10). Additionally, the redox peaks remain defined across a range of scan rates from 0.2 to 1.0 mV s⁻¹ (Fig. 2b), with only minor shifts attributed to kinetic limitations rather than irreversible degradation. The first possibility of such significant improvement is the increased structural stability of PANI by OTf⁻ doping. We performed DFT calculations to demonstrate the structural stability with both anion doping. As illustrated in Fig. 2d, the HOMO–LUMO energy gap (ΔE) of the $SO_4^{2-}$ doping is 2.33 eV with a HOMO energy level of −6.43 eV. The OTf⁻ doping renders a slightly larger energy gap (2.44 eV) and a HOMO level of −6.51 eV. The intrinsic thermodynamic resistance to overoxidation

only increases slightly (80 mV), which is not expected to cause any significant divergence in cycling stability. Another characteristic of OTf⁻ anion is that the hydration shell is disrupted, effectively reducing the water-rich environment in the electrolyte. Meanwhile, OTf⁻ anions are expected to keep water molecules outside the PANI backbone (Supplementary Fig. 12). In-operando Raman spectra (Fig. 2c) show that the onset of strong −OH stretching bands is delayed to approximately 1.5 V, confirming a reduced interaction between water molecules and the polymer. Corresponding changes in the PANI backbone are also delayed until 1.5 V. After 100 cycles, PANI still retains its electrochemical activity, as shown by the dynamic changes in C–N/C=N, C–N⁺/C=N⁺, C=O/C–O, and −OH (Supplementary Fig. 13). Similarly, the polymer backbone exhibits dynamic interaction with OTf⁻ and $Zn^{2+}$ ions after 100 cycles (Supplementary Fig. 14). In a nutshell, the electrochemical stability of PANI largely depends on the hydration of anions, rather than anions themselves.

The hydration shells of anions show two distinct modes of interaction with the polymer backbone. When anions are strongly solvated, desolvation becomes energetically unfavorable, leading to the formation of solvent-separated ion pairs (SSIP), where water molecules remain interposed between the anion and the polymer[34,35]. Conversely, weakly solvated anions can easily shed their hydration shells, forming direct contact ion pairs (CIP) with the polymer backbone (Fig. 3a). By performing in-operando measurements, we are able to reduce noise arising from sample-to-sample variation and from differences between experimental batches. In the case of $ZnSO_4$, which is known for strong solvation with water molecules, we observe a dynamic shift between SSIP and CIP states (Fig. 3b). As the voltage increases, the CIP contribution initially decreases, likely due to the attraction of $SO_4^{2-}$ anions, which draw additional water molecules into the polymer matrix (consistent with DFT prediction), thereby shifting the system toward a

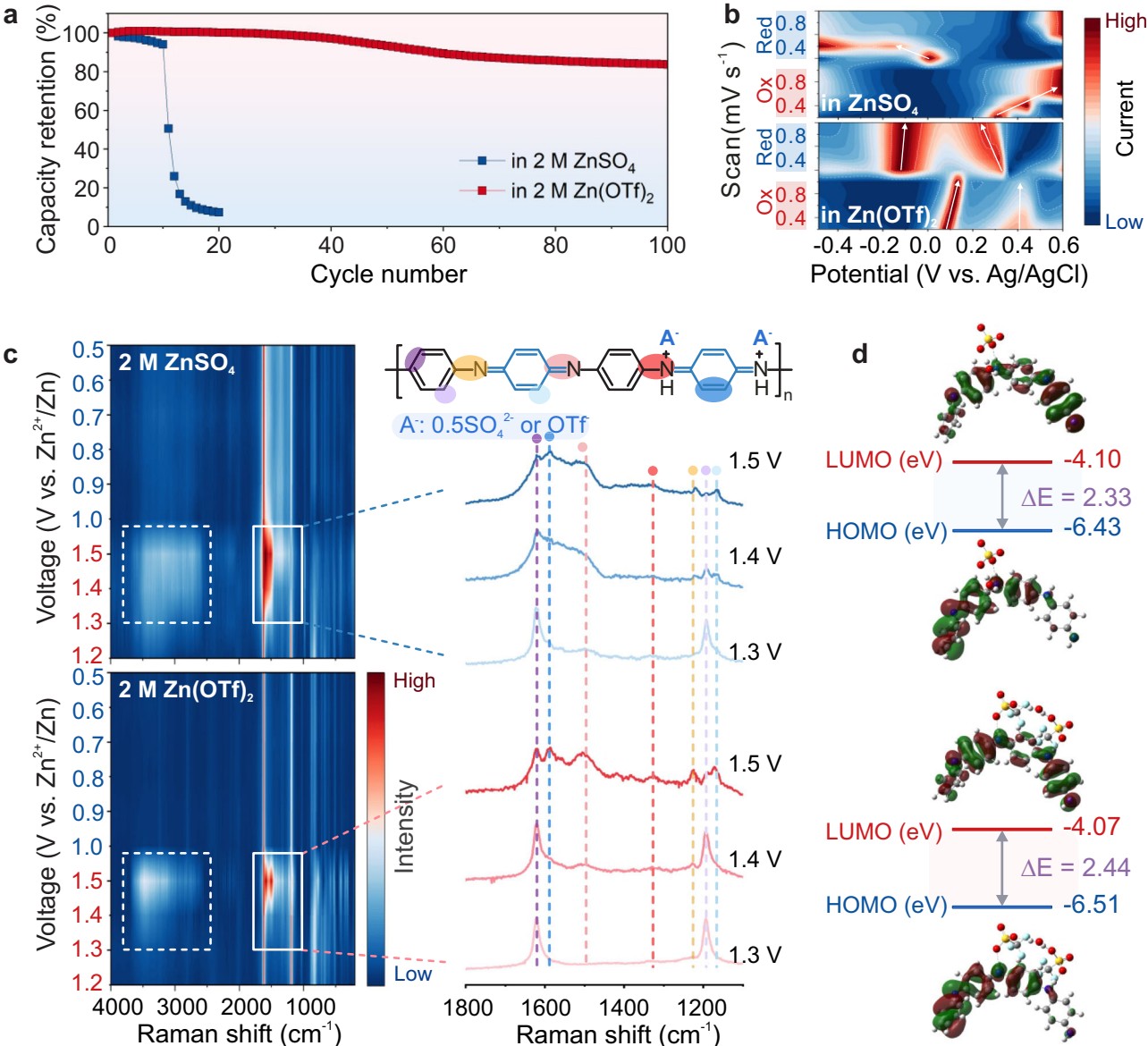

**Fig. 2 | Electrochemical and structural stability of PANI. a** Capacity retention of Zn-PANI microbatteries cycled in 2 M ZnSO$_4$ and 2 M Zn(OTf)$_2$. **b** Contour maps of cyclic voltammetry results for PANI in 2 M ZnSO$_4$ and Zn(OTf)$_2$ at scan rates of 0.2, 0.4, 0.6, 0.8, and 1.0 mV s$^{-1}$. **c** Contour maps of in-operando Raman results of PANI cycled in 2 M ZnSO$_4$ and Zn(OTf)$_2$ electrolytes. The right panel shows the selected regions from 1100 to 1800 cm$^{-1}$. **d** The highest occupied molecular orbital (HOMO) and lowest unoccupied molecular orbital (LUMO) levels and energy gaps (ΔE) for PANI doped by SO$_4^{2-}$ (top) and OTf$^-$ (bottom).

more SSIP configuration. At higher voltages, however, water involved reaction at the polymer backbone starts, leading to the reformation of the CIP state as solvating water is consumed. In contrast, the OTf$^-$ anion, with its disrupted water solvation shell, establishes a more stable interaction with the PANI backbone. This interaction remains largely unaffected by changes in applied voltage, with negligible fluctuation in the surrounding solvent environment (Fig. 3b).

Figure 3c presents the water molecule states recorded by in-operando Raman spectroscopy at the electrode–electrolyte interface under varying voltages. Based on water peak fitting (Supplementary Fig. 15), we classify water molecules into two states based on their hydrogen-bonding network: a low-entropy state, in which water molecules are structured and hydrogen-bonded, and a high-entropy state, where water molecules remain unbound and dynamically disordered[36–38]. In ZnSO$_4$ electrolyte, a sharp increase in the ratio of low-entropy to high-entropy water is observed at 1.4 V, aligning with the transition to a SSIP configuration identified in Fig. 3b. These results

suggest a large uptake of solvated water into the PANI backbone, increasing structural disorder and facilitating unwanted side reactions. Between 1.4 and 1.5 V, the ratio drops, indicating the rapid consumption of solvated water that produces OH$^-$ ions able to deprotonate the PANI backbone, ultimately leading to hydrolytic degradation of PANI. On the contrary, the OTf$^-$-based electrolyte exhibits a stable water profile across 1.2–1.5 V. The consistent ratio of low-entropy to high-entropy water implies a steady interfacial water structure and minimal disruption to the polymer matrix, which also aligns with the stable anion structure shown in Fig. 3b.

The quantification of water uptake was investigated by electrochemical quartz crystal microbalance (EQCM). We begin the measurements after the initial activation phase and monitor the mass changes of PANI during charging and discharging to quantify water uptake (Supplementary Fig. 16). We perform continuous charging and discharging between –0.5 and 0.6 V vs. Ag/AgCl with a scan rate of 20 mV s$^{-1}$ (Fig. 3d). PANI charged in the Zn(OTf)$_2$ electrolyte shows a

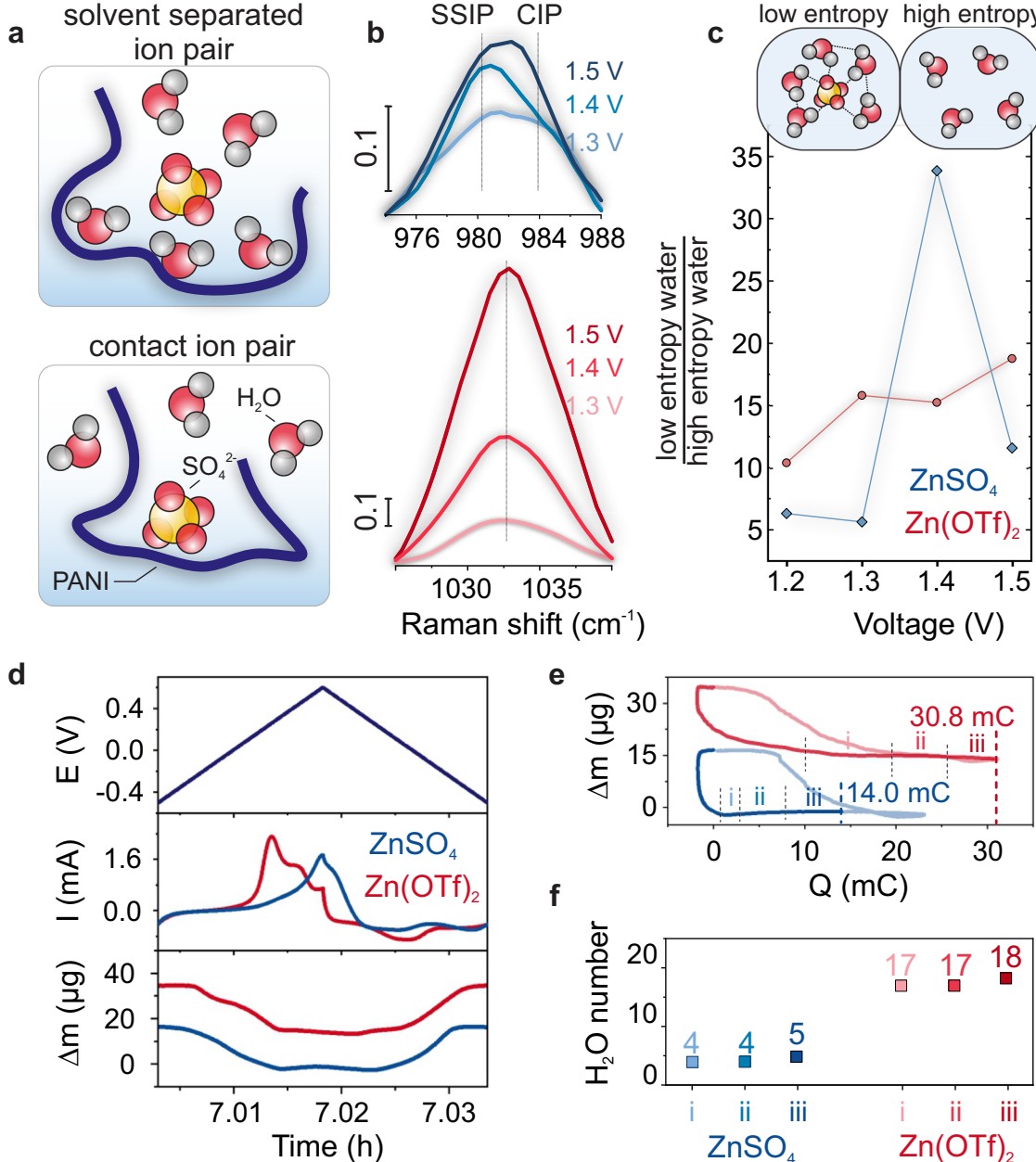

**Fig. 3 | Water interaction with PANI. a** Schematic illustration of solvent-separated ion pair (SSIP) and contact ion pair (CIP) with the polymer backbone. **b** Evolution of Raman peaks for $SO_4^{2-}$ (top) and OTf⁻ (bottom) during oxidation from 1.3 to 1.5 V, relative to the spectra at 1.2 V. **c** Evolution of the ratio between low-entropy water (structured, hydrogen-bonded) and high-entropy water (disordered), extracted from fitted water bands in the in-operando Raman spectra. **d** EQCM results of PANI in 2 M $ZnSO_4$ (blue) and 2 M $Zn(OTf)_2$ (red) under a cyclic voltammetry scan (E (V) vs. Ag/AgCl, scan rate: 20 mV s⁻¹). **e** Mass changes of PANI as a function of charge storage and release in 2 M $ZnSO_4$ and $Zn(OTf)_2$. **f** $H_2O$ numbers per two-electron transfer calculated from **e** during the oxidation process.

higher capacitance (30.8 mC) compared to that in the $ZnSO_4$ electrolyte (14.0 mC), as shown in Fig. 3e. Specifically, a mass increase is detected during the discharging cycle (assigned to the uptake of cations). Interestingly, the mass decreases during the charging, when the polymer uptakes anions, the mass should increase intuitively. This unusual mass reduction suggests the loss of water in the polymer chain. To elucidate the water behavior during anion insertion, the oxidation process was divided into three potential regions: 1.2–1.3 V, 1.3–1.4 V, and 1.4–1.5 V vs. $Zn^{2+}/Zn$. Frequency shifts recorded by EQCM were converted to mass changes using a sensitivity factor of 76.23 Hz μg⁻¹, calibrated by Cu deposition (Supplementary Fig. 17). PANI in $ZnSO_4$ repels 4–5 water molecules across three oxidation stages, while in $Zn(OTf)_2$ it repels 17–18 water molecules (Fig. 3f). The

quantified water withdrawal directly reveals the irreversibility of PANI due to hydrolytic degradation resulting from strong water uptake. Moreover, despite the potential surface degradation suggested by the Raman result (Fig. 2c), strong water withdrawal in $Zn(OTf)_2$ in the range of 1.4–1.5 V indicates the suppressed hydrolytic degradation. The mass of the PANI film remained stable during the rest period in $Zn(OTf)_2$ electrolyte (Supplementary Fig. 18). Conversely, a significant mass loss was recorded in $ZnSO_4$, suggesting that the PANI structure is disrupted, unable to host ions after multiple cycles.

With water-induced degradation successfully mitigated, we proceeded to fabricate a dual-cell microbattery using the micro-origami assembly process shown in Fig. 1d. The final folded structure, shown in Fig. 4a, measures 750 × 750 μm². This compact architecture enables

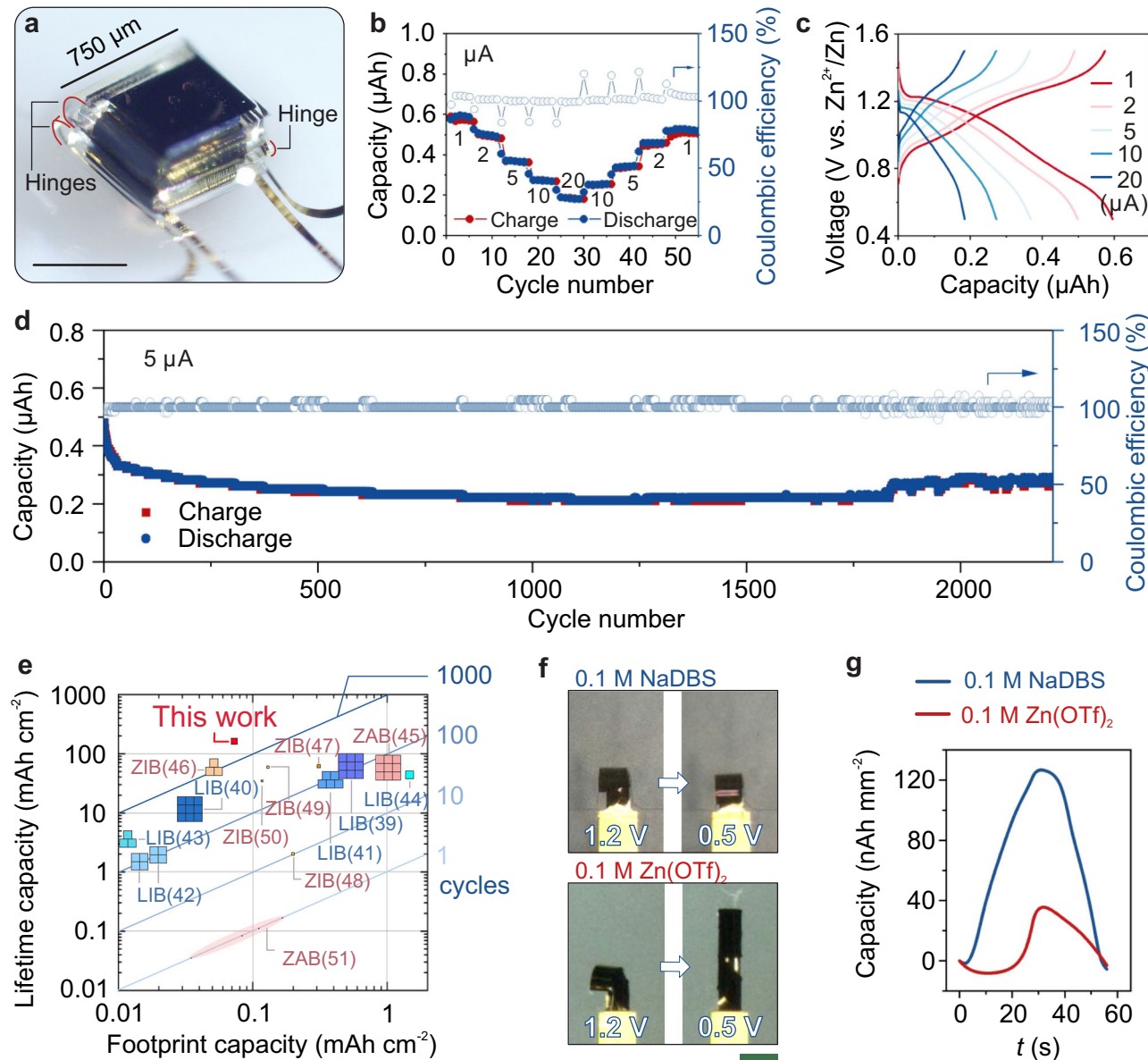

**Fig. 4 | Energy storage performance of PANI microbattery and actuation performance of PPy actuator. a** Optical image of the dual-cell microbattery. Scale bar: 500 μm. **b** Rate performance and **c** galvanostatic charge-discharge profiles of the dual-cell microbattery at currents from 1 to 20 μA. **d** Stability of the dual-cell microbattery cycled at 5 μA. **e** Comparison of microbattery performance with a footprint smaller than 10 mm²[57–69]. One square represents 1 mm². For footprints smaller than 1 mm², the square size is proportionally reduced. ZIB zinc-ion battery, ZAB zinc-air battery, LIB lithium-ion battery. **f** Comparison of PPy actuation in 0.1 M NaDBS and Zn(OTf)₂. Scale bar: 250 μm. **g** Capacity consumption of PPy in 0.1 M NaDBS and Zn(OTf)₂.

two stacked electrochemical cells within a sub-millimeter footprint (0.56 mm²). All electrochemical tests were conducted in PDMS molds to confine and precisely control the electrolyte volume, thereby avoiding performance variations arising from differences in electrolyte amount (Supplementary Note 6, Supplementary Fig. 19). The rate performance of the microbattery is presented in Fig. 4b. At a discharge current of 1 μA, the microbattery delivers an actual capacity of 0.59 μAh (1.05 μAh mm⁻²), exhibiting a clear discharge plateau near 1.23 V in the galvanostatic charge-discharge profiles (Fig. 4c), consistent with the redox features identified in the cyclic voltammetry curves (Fig. 2b). As the current increases, the capacity declines due to kinetic limitations, accompanied by a gradual lowering of the discharge plateau (Fig. 4c). Even at a high current of 20 μA, the device still retains a capacity of 0.19 μAh (0.34 μAh mm⁻²). Notably, the discharge plateau remains visible at around 1.14 V, indicating that redox activity is still sustained at higher rates. Upon reducing the current back to 1 μA, the capacity

recovers to 0.53 μAh (0.95 μAh mm⁻²), demonstrating excellent reversibility with a retention of 90% compared to the initial cycle. Long-term cycling at 5 μA further confirms the device stability: over 2200 cycles are achieved with nearly 100% Coulombic efficiency (Fig. 4d). The initial capacity decay observed in both single-cell and dual-cell configurations is attributed to the re-equilibration of PANI after exposure to NaOH during the self-folding process (Supplementary Note 7, Supplementary Figs. 20–22).

The dual-cell microbattery delivers a lifetime capacity of 161 mAh cm⁻², surpassing previously reported microbatteries less than 10 mm², which generally exhibit lifetime capacities below 100 mAh cm⁻². Moreover, while cycling beyond 1000 cycles is demonstrated in devices larger than 3 mm², where excess active material can buffer degradation, the dual-cell microbattery achieves high cycling stability at a substantially smaller scale, marking a significant step forward in high-density, durable microscale energy storage.

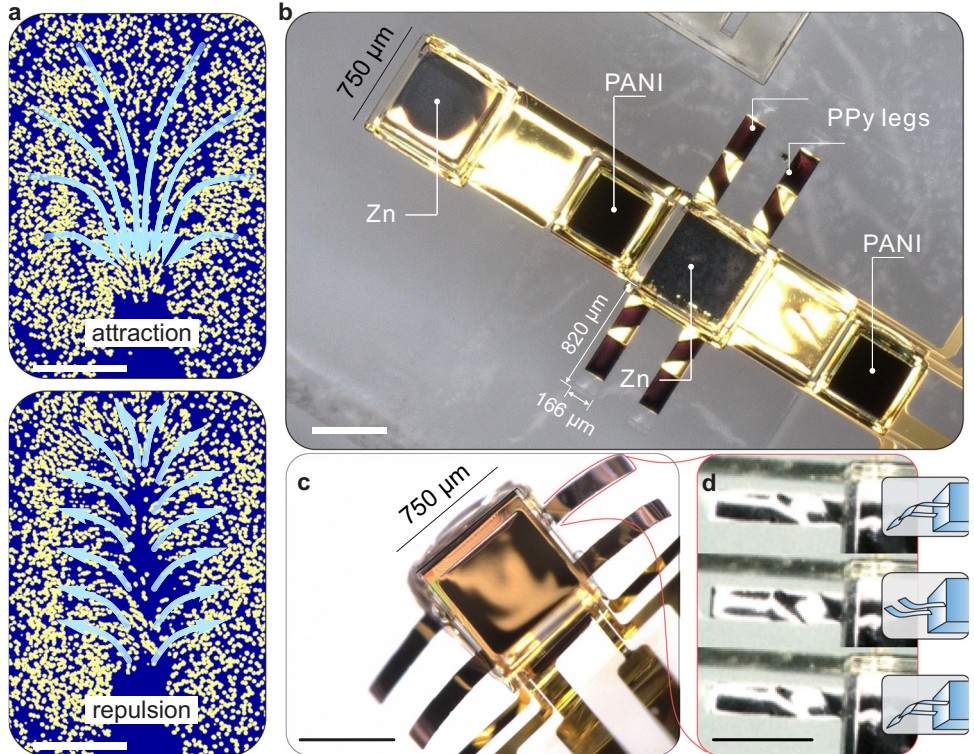

**Fig. 5 | Monolithic integration of the dual-cell microbattery and actuators.**
**a** Flow patterns generated by the PPy micro-actuator driven by a dual-cell micro-battery. Scale bars: 250 μm. Yellow particles: 3-μm fluorescent microspheres. **b** Optical image of the precursor thin films consisting of two sub-batteries and four PPy actuators, which self-folds into **c** a monolithically integrated 3D device. Each leg measures 820 μm long, 166 μm wide. Scale bars: 500 μm. **d** Optical images of flapping motions of PPy oscillators driven by the local microbattery. Scale bar: 500 μm.

The influence of water extends beyond energy storage in conjugated polymers, and it also plays a pivotal role in enabling actuation. Swelling and deswelling of the polymer backbone, driven by the insertion and extraction of ions and water molecules, enable mechanical deformation in polymer actuators. This mechanism has made polypyrrole (PPy) one of the most widely used electroactive polymers in applications ranging from artificial muscles to micromechanical systems[4,7,22,39,40]. Furthermore, PPy will not be damaged by exposure to NaOH during the self-folding process (Supplementary Fig. 23). The efficiency of actuation is tightly coupled to the speed at which water molecules and ions can be exchanged within the polymer. To examine this relationship, we compared the actuation performance of PPy films in two electrolyte systems: sodium dodecylbenzenesulfonate (NaDBS) and $Zn(OTf)_2$. Both electrolyte allows PPy to cycle for 500 times (Supplementary Fig. 24). NaDBS is a benchmark electrolyte for PPy actuators due to its mechanical stability and long-term durability[41–43]. The strong actuation is attributed to the high water exchange capacity of up to 401 water molecules (Supplementary Fig. 25). However, the PPy film in $Zn(OTf)_2$ quickly returns to a flat state under 0.5 V (Fig. 4f), whereas the film in NaDBS retains a curved configuration, indicating slower relaxation dynamics. Specifically, the slow relaxation in NaDBS is due to a two-step recovery process (Supplementary Fig. 25). As PPy reduces, the mass increases due to the large amount of water intercalation while the polymer backbone needs more time to accommodate water molecules, resulting in a decrease in mass to achieve equilibrium. In contrast, the lower hydration in $Zn(OTf)_2$ allows for continuous intercalation of a small number of water molecules, thus providing a rapid relaxation. This quick mechanical response demonstrates the superior actuation efficiency of $Zn(OTf)_2$. The reduced but continuous water exchange in $Zn(OTf)_2$ leads to a much lower energy requirement for the actuation process, decreasing approximately 4-fold when compared to NaDBS,

as shown in Fig. 4g. This ultralow power consumption is critical for miniaturized and autonomous systems. This suggests that efficient water exchange, enabled by the disrupted solvation environment in $Zn(OTf)_2$, plays a key role in amplifying actuation speed and reducing energy demand. Furthermore, the similarly fast relaxation observed for the PPy actuator in NaOTf solution confirms that the dominant effect arises from the OTf⁻ anion rather than the cation (Supplementary Note 8, Supplementary Fig. 26).

The efficient actuation achieved with PPy allows for cilia-like motion, a strategy widely exploited in nature for microscale fluid manipulation[44–46]. The PPy micro-actuator is able to manipulate microspheres suspended in an aqueous environment. When charged by the microbattery, the PPy structure rolls upward, drawing microspheres together and causing attraction (Fig. 5a, upper panel). Upon discharging, the PPy unrolls, releasing the stored mechanical tension and dispersing the microspheres back into the surrounding fluid (Fig. 5a, bottom panel). This dynamic and reversible movement showcases the promise of integrated microsystems for fluidic control at the microscale, which is therefore fabricated through a self-folding process that transforms a planar precursor structure into a three-dimensional form (Fig. 5b). To maximize energy density within a limited footprint, we employed a dual-cell microbattery design, where one planar electrode (specifically, the bottom Zn electrode) was integrated with four PPy legs for actuation, which are powered by a localized energy source (Fig. 5c). As demonstrated in Fig. 5d, the PPy legs, driven by the dual-cell microbattery, exhibit robust flapping motions, bending downward upon charging and returning upward upon discharging. An external manual switch circuit (Supplementary Fig. 1) is connected externally between the dual-cell microbattery and actuators. Manually changing the connection of actuators to the positive and negative terminals of the microbattery allows for repeated bending and relaxation. The repeated motion sequence is provided in the

Supplementary Movie 2. Additionally, the dual-cell microbattery can individually power a digital watch (Supplementary Fig. 27). The on-chip circuit that connects two dual-cell microbatteries in series to meet the voltage demand enables the powering of green and red LEDs, demonstrating the potential for integration in multifunctional microsystems (Supplementary Fig. 28).

## Discussion

In summary, this study demonstrates that water plays a key role in the stability and function of conjugated polymer thin films. Extensive water absorption by the conjugated polymer enhances actuation; however, it also increases energy consumption and slows down the relaxation process due to the transition from excessive water uptake to an equilibrium state. Additionally, water within the polymer backbone facilitates hydrolytic degradation when used as an energy storage device. By reducing the water content within the polymer backbone, the electrochemical stability of the conjugated polymers can be significantly improved, and the electrochemical response during actuation can be accelerated, thus lowering power consumption. This understanding allows for the development of a monolithically integrated energy storage (0.56 mm$^2$) and actuation system, achieving a lifetime capacity of 161 mAh cm$^{-2}$, fluid manipulation similar to cilia driven by the local microbattery, and power sources for electronic components. Moreover, the role of water in the electrochemical behavior of conjugated polymers extends beyond PANI and PPy to other systems such as PEDOT, where regulation of anion hydration markedly improves electrochemical reversibility (Supplementary Note 9, Supplementary Fig. 29). Importantly, this hydration-control strategy is not limited to OTf; other weakly hydrated anions similarly enhance conjugated polymer performance, even in more physiologically relevant media (Supplementary Note 10, Supplementary Figs. 30 and 31). Pioneering studies have already demonstrated the promise of microscale batteries for neuromodulation and bioresorbable pacemakers[47,48], establishing the feasibility and biomedical relevance of on-board power at extreme scales. Building on these advances, our work shifts the focus from function to foundation, showing how control over electrolyte–polymer interactions in aqueous environments can be used to endow electrochemical materials with both stability and multifunctionality. These insights point toward untethered microscale systems in which energy storage and actuation are no longer separate components, but chemically integrated elements of implantable bioelectronics and microrobots.

## Methods

### Photosensitive polymer layers (SL/HG/PI)

Polymer solutions were prepared as follows. Briefly, The sacrificial layer (SL) precursor was prepared from a lanthanum–acrylate coordination polymer[49]. Acrylic acid (AA, Alfa Aesar) and hydrated lanthanum(III) chloride (Thermo Scientific, 99%) were mixed in deionized water (10 g AA and 4.86 g LaCl$_3$), yielding a lanthanum acrylate precipitate. The precipitate was collected by filtration and dried at 40 °C for 10 h. The dried material was redissolved in acrylic acid to obtain a 25 wt% solution and photosensitized with 2 wt% 2-benzyl-2-(dimethylamino)-4-morpholinobutyrophenone (DBMP, TCI Chemicals, >98%) and 3 wt% methyl diethanolamine.

The hydrogel (HG) precursor was prepared by reacting N-(2-hydroxyethyl)acrylamide (HEAA) with poly(ethylene-alt-maleic anhydride) (PEMA, Sigma-Aldrich, Mw 100,000–500,000) in N,N-dimethylacetamide (DMAc, Thermo Scientific, 99.5%)[49]. Briefly, 6 g of PEMA was dissolved in 50 mL DMAc, followed by addition of 5.75 g HEAA. The reaction was carried out at room temperature for 10 h. The solution was subsequently photosensitized with 2 wt% DBMP.

The polyimide (PI) layer was prepared via a photosensitive polyamic acid (PAA) precursor[49]. PAA was synthesized by polycondensation of 3,3′,4,4′-benzophenonetetracarboxylic dianhydride (BPDA, Alfa Aesar, 97%) and 3,3′-diaminodiphenylsulfone (DADPS, Fisher Scientific) in DMAc. Specifically, 9.93 g DADPS was dissolved in 20 mL DMAc, followed by addition of 12.8 g BPDA, and the mixture was stirred at 70 °C for 12 h. The resulting PAA solution was modified by neutralization with 12.5 g 2-dimethylaminoethyl methacrylate (Alfa Aesar, 97%) and photosensitized with 2 wt% DBMP.

### Fabrication of self-folding platform

The origami-inspired dual-cell microbatteries were fabricated based on a patternable and stimuli-responsive polymeric platform[50]. Briefly, glass and silicon wafers (22 × 22 mm$^2$) served as substrates, which were cleaned with acetone and isopropanol, followed by oxygen plasma treatment (TEPLA, 200 W, 2 min). The SL was spin-coated at 3000 rpm for 35 s and soft-baked at 35 °C for 10 min to remove residual solvent, and then patterned via photolithography using a Karl Süss MA6 mask aligner. Ultraviolet exposure (365 nm, 4 mJ cm$^{-2}$, 60 s) defined the pattern, after which unexposed regions were removed by immersing the substrate in deionized water for 10 s and rinsing with (1-methoxy-2-propyl) acetate (Fisher Scientific) for 30 s. The patterned SL was thermally stabilized at 220 °C for 10 min. Subsequently, the HG layer was spin-coated at 4000 rpm for 35 s, soft-baked at 40 °C for 10 min, and then exposed for 50 s. Development was carried out in diethylene glycol monoethyl ether (DEGMEE, Sigma-Aldrich) for 2 min, followed by rinsing in (1-methoxy-2-propyl) acetate for 30 s, and thermal stabilization at 220 °C for 10 min. The PI layer was then spin-coated at 3000 rpm for 35 s, soft-baked at 50 °C for 10 min and exposed for 90 s. Development was performed in a solvent mixture of N-ethyl pyrrolidone (Apollo Scientific), DEGMEE, and ethanol (VWR chemicals) (volume ratio 4:2:1) for 2 min, followed by rinsing in (1-methoxy-2-propyl) acetate for 30 s and baking at 220 °C for 10 min. The final thicknesses of the SL, HG, and PI layers were approximately 300 nm, 1000 nm, and 1000 nm, respectively.

Ti (10 nm)/Au (50 nm) metal layers, as current collectors, were photolithographically patterned using commercial photoresist AZ5214E (MicroChemicals GmbH) exposed with a Heidelberg MLA100 Maskless Aligner. Metal deposition was conducted in an electron beam evaporator (Creavac) at a rate of 0.5 Å s$^{-1}$, and excess metal was removed by standard lift-off procedure.

A ~900 nm thick SU-8 layer was patterned to serve as a passivation layer, with openings at the electrode regions to allow for Zn anode and polyaniline (PANI) cathode deposition. SU-8 3005 (Kayaku) was patterned to form ~140 μm-thick frames for the anode and cathode. All SU-8 layers were developed using mr-Dev 600 (Micro Resist Technology).

The self-folding process started with the etching of the SL in a 3.7 wt% HCl solution (Technic, 37%). Subsequently, the freestanding device was rinsed with DI water and moved to a 0.1 M NaOH (Sigma-Aldrich) solution. The device folded autonomously due to the swelling of the HG hinges in the alkaline solution.

### Zn and PANI electrodeposition

The PANI cathode was deposited at a constant voltage of 1.0 V for 60 s in a three-electrode setup with an Ag/AgCl (3.5 M KCl) reference electrode and a carbon cloth as the counter electrode. The deposition solution consisted of 0.5 M aniline monomer (Sigma-Aldrich) and 1 M H$_2$SO$_4$ (Sigma-Aldrich, 98%). The Zn anode was deposited using a constant current of 100 μA for 10 min in a two-electrode setup with a Zn plate as the counter electrode and 2 M ZnSO$_4$ (Sigma-Aldrich) as the electrolyte.

### Battery performance tests

Cyclic voltammetry (CV) curves were recorded using an electrochemical workstation (Metrohm Autolab MULTIAUTOLAB/M101). Linear sweep voltammetry (LSV) measurements were conducted in a three-electrode configuration at a scan rate of 5 mV s$^{-1}$, using a Pt mesh counter electrode, stainless-steel working electrode, and an Ag/AgCl reference electrode. Galvanostatic cycling of Zn‖Zn symmetric cells

was performed in a coin-cell configuration to compare their electrochemical performance in 2 M $ZnSO_4$ and 2 M $Zn(OTf)_2$ (Sigma-Aldrich) electrolytes. Both the galvanostatic charge/discharge profiles and the rate performance were measured by a battery testing system (Biologic BCS-805). The battery was charged and discharged at a constant current within a voltage range of 0.5–1.5 V. Electrochemical quartz crystal microbalance (EQCM, a Gamry eQCM 15 M instrument using an Au-coated 5 MHz quartz sensor with an electroactive area of 0.785 $cm^2$) and CV test were measured simultaneously in the same three-electrode configuration to investigate the in-situ mass changes of PANI during its redox reactions.

The mass change was calculated using the Sauerbrey equation (Eq. (1))[51]:

$$\Delta f = - C_f \Delta m \tag{1}$$

$\Delta m$ – Mass change of the electrode (g)
$\Delta f$ – Frequency change (Hz)
$C_f$ – sensitivity factor (Hz/g)

The sensitivity factor $C_f$ was calibrated by Cu deposition experiment (in Supplementary Fig. 17). The $C_f$ was calculated to be 76.23 Hz $\mu g^{-1}$. In Fig. 3f, the calculation of water numbers in the oxidation process of PANI is based on the slopes ($\Delta m/\Delta Q$) in Fig. 3e. For example, when PANI is charged in 2 M $ZnSO_4$, from 8 to 14 mC corresponding to the oxidation domain (iii), the slope (mass change per electron) $\Delta m/\Delta Q$ is 28.08 $\mu g\ C^{-1}$, which is further converted using Eq. (2), yielding an apparent molar mass per transferred electron ($M/z$) of 2.71 g $mol^{-1}$.

$$M/z = F \Delta m/\Delta Q \tag{2}$$

Here, $\Delta Q$ is the charge transferred during the electrochemical process (in coulombs), $F$ is the Faraday constant (96485 C $mol^{-1}$). Taking the two-electron transfer reaction in PANI as a reference, this region involves the insertion of $SO_4^{2-}$, so the calculated number of water molecules according to Eq. (3) is −5.0.

$$Water number = \frac{2*\frac{M}{z} - 96}{18} \tag{3}$$

The negative number means that 5 water molecules move out from PANI.

### Electropolymerization of polypyrrole (PPy) actuators and the actuation control

PPy actuators were deposited at a constant voltage of 0.8 V for 30 s in a three-electrode setup with the Ag/AgCl (3.5 M KCl) reference electrode and a Pt foil as the counter electrode. An aqueous monomer solution (0.1 M pyrrole (Sigma-Aldrich, 98%) and 0.1 M sodium dodecylbenzene sulfonate (NaDBS, Thermo Scientific)) was used as the electrolyte for PPy electrodeposition. A switch circuit (Supplementary Fig. 1) is used to realize the repeated bending and relaxation of actuators. Specifically, when switches S2 and S3 are closed, the actuator is injected with ions, inducing bending. Subsequently, disconnecting S2 and S3, while closing switches S1 and S4, reverses the polarity of the actuator, leading to the release of ions and consequently relaxation.

### Electropolymerization of PEDOT

PEDOT was deposited at 1.25 V for 60 seconds in a three-electrode configuration using an Ag/AgCl (3.5 M KCl) electrode as the reference and a Pt mesh as the counter electrode. The deposition electrolyte consisted of 0.01 M 3,4-ethylenedioxythiophene (EDOT, Sigma-Aldrich) monomer dissolved in 2 M $Zn(OTf)_2$.

### Material characterizations

The morphologies were characterized by scanning electron microscopy (SEM, GAIA3 TESCAN). Raman spectroscopy was conducted at room temperature using a LabRAM HR Evolution spectrometer (Horiba) with a 458 nm excitation laser. X-ray photoelectron spectroscopy (XPS) analyses were conducted using an ESCALAB 250Xi spectrometer (Thermo Fisher Scientific, K-Alpha series), equipped with a monochromatic Al Kα radiation source. Binding energies were calibrated using the C 1$s$ peak at 284.8 eV.

### Density functional theory (DFT) calculation

All the first-principles calculations were carried out by Gaussian software package[52]. Geometry optimizations and vibrational frequency analyses were performed using the hybrid B3LYP functional[53] in combination with the 6–311++G(d,p) basis set for all atoms. To accurately account for long-range weak interactions, Grimme's D3 dispersion correction with Becke–Johnson damping (DFT-D3(BJ)) was employed[54,55]. To better simulate the water-based battery environment, the Solvation Model based on Density (SMD) implicit solvent model was used for all the calculations[56]. In Eq. (4), the binding energy was defined as

$$E_b = E_{PANI@anion@H_2O} - \left( E_{PANI@anion} + E_{H_2O} \right) \tag{4}$$

where anion = $SO_4^{2-}$ or $2OTf^-$, and all energies were computed consistently with the same solvation model.

The electronic properties of the PANI-based systems were analyzed by evaluating the frontier molecular orbital energies obtained from the optimized geometries. In Eq. (5), the energy gap ($\Delta E$) was defined as the difference between the lowest unoccupied molecular orbital (LUMO) and the highest occupied molecular orbital (HOMO) energies:

$$\Delta E = E_{LUMO} - E_{HOMO} \tag{5}$$

where $E_{HOMO}$ and $E_{LUMO}$ are the Kohn–Sham orbital energies extracted from the DFT calculations. The spatial distributions of the HOMO and LUMO were visualized to elucidate the charge density distribution and to provide insights into the electron transfer characteristics induced by different anions.

## Data availability

The data generated in this study are provided in the Supplementary Information/Source Data file. Source data are provided with this paper. All data are available from the corresponding author upon request. Source data are provided with this paper.

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

## Acknowledgements

M.Z. acknowledges the support of the European Union (ERC, SMAD-BINS, 101039802) and German Research Foundation DFG (ZH 989/2-1). O.G.S. acknowledges financial support from the Leibniz Program of the German Research Foundation (no. SCHM 1298/26-1). L.M. acknowledges the support from the Brazilian National Council for Scientific and Technological Development through the research productivity program (CNPq 312243/2025-1). D.K. acknowledges the support of the German Research Foundation DFG (KA 5051/3-1). W.Z. acknowledges the scholarship support from the China Scholarship Council. The authors acknowledge Q. Guo for assistance with EQCM training. The authors acknowledge M. Yu, C. Schmidt, and A. Dumler for technical support. The authors acknowledge A.-M. Placht and E. Auerswald for the SEM measurement.

## Author contributions

M.Z. and O.G.S. conceived the idea and supervised the project. W.Z. designed and carried out most of the experiments. L.M. assisted with PPy deposition and actuator discussions. J.M. performed the DFT calculations. C.B., D.K., D.D.K., and A.I.E. contributed to the micro-origami design. J.Q. conducted the Raman measurements. Y.H. performed the XPS measurements. H.T., L.M.M.F., and Y.Y. assisted in experimental design and manuscript preparation. Y.L. and V.K.B. supported the LED tests. All authors discussed the results and contributed to the manuscript.

## Funding

## Competing interests

The authors declare no competing interests.
