## [Peer Review file · Nature Communications]

A Bioinspired Microdevice Unifying Energy Storage and Actuation Through Hydration Control

Corresponding Author: Dr Minshen Zhu

Version 0:

Reviewer comments:

Reviewer #1

(Remarks to the Author)

This manuscript presents a compelling and innovative approach to integrating energy storage and actuation into a single, monolithic conjugated-polymer-based platform. The authors address a long-standing challenge of balancing energy density, actuation performance, and device longevity by studying the critical role of water in polymer degradation and dynamic response. The work is both mechanistically insightful and technologically impactful, representing an important step toward bio-inspired, untethered microsystems for robotics and bioelectronics. This is a strong and highly original contribution that is likely to attract wide interest. I recommend acceptance after revision.

(1) The folding is triggered in 0.1 NaOH, would this exposure degrade battery/actuator materials?

(2) The authors compared the actuation performance of PPy in NaDBS and Zn(OTf)₂, and realized that Zn(OTf)₂ provides a rapid relaxation. Would this apply to other solvent containing (OTf)⁻, such as Na(OTf)?

(3) I would suggest the authors compare the cycling stability of the actuators in NaDBS and Zn(OTf)₂.

(4) Much of the mechanism is demonstrated on PANI for storage and PPy for actuation. Is this generally applicable to other conjugated polymers? It could be useful to add data on a third polymer system.

Reviewer #2

(Remarks to the Author)

In this work, the authors developed a sub-millimeter monolithic device that unifies energy storage and actuation in a single, synergistic platform. Through hydration control, the performance of both micro-battery and actuator has been improved, providing a representative example using multifunctional materials to construct integrated devices. Therefore, it could be recommended for publication in Nature Communications. Before, some issues still need to be addressed.

1. The long-term cycle performance should be evaluated for the single micro-battery. Further, the reason for rapid capacity degradation of dual-cell micro-battery in initial cycles (Figure 4d) should be explained. In addition, at the current of 5 μ A, why is the capacity of dual-cell micro-battery in Figure 4b and 4d different?

2. The areal capacity of single cell should be provided, not just a lifetime capacity.

3. In a very small space, the precise addition and confinement is usually a challenge. The authors should introduce how to realize this in detail.

4. The linear sweep voltammetry should be provided for the two electrolyte, to support whether the water decomposition happen at higher voltage.

5. The PANI cathode shows obvious performance difference in two electrolyte. How is the Zn anode? Does the Zn anode give different reversibility in the two electrolyte? Would Zn anode influence the cycle of micro-battery?

Reviewer #3

(Remarks to the Author)

Reviewer comments:

In this paper, Zhang et al. developed a sub-millimeter monolithic device that unifies energy storage and simple actuation functions. Importantly, the authors innovatively used triflate (OTf⁻) to mitigate the hydration issue of the conjugated polymer polyaniline (PANI), thereby improving its cycling stability and device lifetime. A thorough analysis based on in-operando Raman spectroscopy, time-resolved mass spectrometry, and various simulation approaches has been conducted to elucidate the mechanism and effects of water in conjugated polymers. Further, the developed dual-cell and multiple-cell microbatteries have shown great potential in the important areas of miniature batteries and microrobots. Therefore, I would recommend the publication of this work.

Before acceptance, there are some major issues that need to be addressed.

Major Issues:

1. Figure 1 should be significantly revised. Here are my concerns and suggestions. The main innovation of addressing PANI hydration and degradation is not shown. Fig. 1a vaguely explains the mechanism of the biological muscle fascicle and is not directly connected to the device structure. I suggest moving it to the SI. The battery reactions and actuator mechanisms are not shown, which could be unclear to non-specialist readers. The device's internal structure after folding is unclear to me. How the battery is connected to the actuators is unclear.
2. It is also unclear to me why the authors focused on a dual-cell microbattery design, when they could hugely increase the cell numbers, such as a five-cell microbattery shown in Fig. S3. Other structural dimensions are also vaguely explained, e.g., 820 μm long and 166 μm wide. I wonder what the influence of battery size is? What is the minimal size the authors can achieve, while keeping the rationale of the PANI device valid?
3. By replacing sulphate anions with triflate, PANI shows a significant improvement in its stability. What is the aqueous environment for this test? What is the device encapsulation? How will the device perform in physiological environments?
4. The shifts of Raman peaks in Fig. 3b seem marginal. How valid is the methodology to differentiate SSIP and CIP states?
5. The final application in driving actuation is interesting. However, I cannot see a strong relevance to muscle fascicles (see my comment 1). The one-off actuation and the use of a dual-cell design, instead of others, should be further explained.
6. Recent work has shown the importance of solid-state and soft microbatteries in driving ionic movements and various biological modulation, as well as potential applications in microrobots. The authors are recommended to discuss more about the potential applications of their microbatteries by comparison with other work.

Reviewer #4

(Remarks to the Author)

This manuscript reports a monolithic, sub-millimeter integrated device that combines conjugated-polymer-based energy storage with electrochemically driven actuation (PPy micro-actuators). Using a broad experimental toolkit, the authors propose a design principle: disrupting ion hydration shells reduces water ingress into the conjugated polymer, suppresses PANI hydrolysis, and thereby improves storage stability as well as actuation bandwidth/energy efficiency. Linking solvation-structure engineering to both durability and performance at the sub-millimeter scale is timely and impactful for microsystems that co-localize power and motion (e.g., microrobotics, lab-on-chip). The concept is compelling, experimentally rich, and cross-disciplinary—appropriate for Nature Communications. Overall, this is one of the most beautiful studies I have reviewed in this area, as I know it is always very challenging to integrate battery systems into such small scales while still maintaining decent performance. That said, several issues still should be considered before publication.

1. The manuscript demonstrates application-relevant functions (e.g., LED lighting, powering low-power digital watches), but many of these exciting results are in the Supplementary Information. To improve readability and impact, the authors are suggested to migrate a subset into the main figures—for example, consider integrating current Fig. S2 into Fig. 1f.
2. The folding process is novel and crucial for efficient space utilization, but it remains difficult for me to follow without watching the video. The schematics are likely due to an unhelpful viewing angle. The authors are suggested to add a clearer schematic (e.g., an alternate perspective or step-wise panels with numbered stages and motion arrows) that makes the self-folding mechanism and hinge locations unambiguous.
3. Please add scale bars (and, if feasible, frame-overlay timestamps) to all Supplementary Videos and main figures.
4. The red line in Fig. 4a requires a precise definition in the caption and/or main text (what quantity it represents, how it is derived, and its role in the analysis).

Version 1:

Reviewer comments:

Reviewer #1

(Remarks to the Author)

The authors have answered all my questions and comments. I recommend the acceptance of the manuscript in this version.

Reviewer #2

(Remarks to the Author)

The authors well considered my comments, now it is acceptable for publication without further revision.

Reviewer #3

(Remarks to the Author)

The authors have addressed most of my comments. The revised manuscript is clear, and the development is important in the field of microbatteries. My only remaining concern is that the use of external control/logic circuits to reversibly charge-discharge micro-actuators in Fig 5 should be clearly mentioned in the main text. I would be happy to recommend publication after this revision.

Reviewer #4

(Remarks to the Author)

The authors have addressed all my previous concerns, and thus I recommend their acceptance.

Reply to Reviewers

Reviewer 1:

This manuscript presents a compelling and innovative approach to integrating energy storage and actuation into a single, monolithic conjugated-polymer-based platform. The authors address a long-standing challenge of balancing energy density, actuation performance, and device longevity by studying the critical role of water in polymer degradation and dynamic response. The work is both mechanistically insightful and technologically impactful, representing an important step toward bio-inspired, untethered microsystems for robotics and bioelectronics. This is a strong and highly original contribution that is likely to attract wide interest. I recommend acceptance after revision.

1. The folding is triggered in 0.1 M NaOH, would this exposure degrade battery/actuator materials?

Reply: Thank you for raising this important point. In our initial work, we qualitatively judged that brief exposure to 0.1 M NaOH does not severely damage PANI or PPy, because the assembled sub-millimeter devices still showed good performance. Nevertheless, we agree that this issue deserves quantitative investigation, and we have therefore added new data to evaluate the effect of 0.1 M NaOH.

For PANI, immersion in 0.1 M NaOH increases resistance, as expected due to deprotonation. However, the extent of this increase strongly depends on the electrolyte. In Zn(OTf)₂, the resistance increases by less than a factor of two, whereas in ZnSO₄, it increases by more than twenty times (Figure S3). This clearly indicates that the anion plays an important role in the behavior of PANI in aqueous electrolytes. Consistently, cyclic voltammetry after NaOH treatment shows that PANI in Zn(OTf)₂ remains stable, while in ZnSO₄, the CV shape becomes strongly distorted, indicating that the deprotonation cannot be effectively reversed in ZnSO₄ (Figure S4).

For PPy, the key parameter is the charge required for actuation. We therefore directly measured the charge consumption after immersion in NaOH. In Zn(OTf)₂, the charge consumption remains essentially unchanged (Figure S23). The slight decrease in charge can be attributed to partial deprotonation, which slightly reduces the number of active sites. However, this effect must be considered together with cycling stability. Importantly, PPy can still be operated for

500 cycles after NaOH treatment, confirming that its electrochemical actuation performance is well preserved (Figure S23).

We have added related discussions on pages 5-6 (highlighted texts).

Figure S3. Bode plots of PANI in **a**, 2 M $\text{Zn}(\text{OTf})_2$ and **b**, 2 M ZnSO_4 recorded before and after immersion in NaOH solution.

Figure S4. CV curves of PANI in 2 M $\text{Zn}(\text{OTf})_2$ before (**a**, **b**) and after (**c**, **d**) immersion in NaOH solution. The curves were recorded at a scan rate of 0.2 mV s^{-1} (**a**, **c**) and at scan rates from 0.2 to 0.8 mV s^{-1} (**b**, **d**).

Figure S23. Capacity consumption of PPy in 0.1 M Zn(OTf)₂ before and after immersion in NaOH solution.

Figure S24. Electrochemical cycling behaviour of PPy in 0.1 M Zn(OTf)₂ after immersion in NaOH, measured at a scan rate of 0.1 V s⁻¹.

2. The authors compared the actuation performance of PPy in NaDBS and Zn(OTf)₂, and realized that Zn(OTf)₂ provides a rapid relaxation. Would this apply to other solvent containing OTf, such as NaOTf?

Reply: Thank you for your insightful question. To address this point, we performed additional actuation tests of PPy in 0.1 M NaOTf (Figure S26). Notably, PPy in NaOTf also shows a reversible redox response and rapid relaxation behavior (relaxation at 0.5 V), comparable to that observed in Zn(OTf)₂. This result indicates that fast actuation behavior is not specific to one cation but is strongly associated with the anion environment. We therefore believe that this additional test further supports our interpretation that a triflate-regulated hydration environment, where weakly coordinating triflate anions and their hydrophobic -CF₃ groups

promote interfacial dehydration, facilitates faster ion and solvent reorganization during actuation.

We have added related discussions on pages 13-14 (highlighted texts).

Figure S26. Optical images of PPy actuation in 0.1 M NaOTf. Scale bar: 250 μm .

3. I would suggest the authors compare the cycling stability of the actuators in NaDBS and $\text{Zn}(\text{OTf})_2$.

Reply: Thanks for your constructive suggestion. We have conducted additional experiments to investigate the cycling stability of PPy actuators, as shown in Figure S24. PPy in $\text{Zn}(\text{OTf})_2$ maintains stable cycling for 500 cycles (Figure S24b). In contrast, PPy in NaDBS shows continuous performance degradation and develops pronounced polarization after approximately 100 cycles (Figure S24a). This distinct difference indicates that the suppressed water exchange in $\text{Zn}(\text{OTf})_2$ effectively mitigates PPy degradation during long-term cycling. We have added related discussions on page 13 (highlighted texts).

Figure S24. Electrochemical cycling behaviour of PPy in **a**, 0.1 M NaDBS and **b**, 0.1 M Zn(OTf)₂ after immersion in NaOH, measured at a scan rate of 0.1 V s⁻¹.

4. Much of the mechanism is demonstrated on PANI for storage and PPy for actuation. Is this generally applicable to other conjugated polymers? It could be useful to add data on a third polymer system.

Reply: We sincerely appreciate this excellent question from the reviewer, which encouraged us to consider the mechanism from a broader and more fundamental perspective. To evaluate whether the hydration-controlled mechanism disclosed in this work is generally applicable to conjugated polymers, we introduced a third representative p-type conducting polymer, Poly(3,4-ethylenedioxythiophene) (PEDOT), as an additional model system.

PEDOT is fundamentally different from PANI and PPy in terms of backbone chemistry and redox states, but it shares the same essential electrochemical feature: its doping and dedoping processes are governed by reversible anion insertion and expulsion, making it an ideal platform to test the generality of an electrolyte-regulated hydration mechanism. PEDOT films were prepared by electropolymerization of 3,4-ethylenedioxythiophene monomers (Figure S29a) and evaluated in both 2 M ZnSO₄ and 2 M Zn(OTf)₂ using a three-electrode configuration.

As shown in Figure S29b, PEDOT in ZnSO₄ exhibits broadened and dispersed redox peaks, indicating sluggish and less uniform doping/dedoping processes. In contrast, PEDOT in Zn(OTf)₂ (Figure S29c) shows a much more reversible electrochemical response, with a significantly reduced peak-to-peak separation (≈ 40 mV in Zn(OTf)₂ versus 230 mV in ZnSO₄), reflecting faster kinetics and improved redox reversibility.

PEDOT has a different molecular structure, charge delocalization mode, and redox chemistry from both PANI and PPy. The fact that the same electrolyte dependence is observed across all three systems strongly indicates that the observed behavior is not polymer-specific, but instead originates from a more general electrolyte-controlled hydration and solvation environment. Specifically, the weakly coordinating nature of OTf⁻ and the presence of hydrophobic -CF₃ groups promote interfacial dehydration and facilitate ion/solvent reorganization, thereby universally enhancing the kinetics and reversibility of doping and dedoping processes in conjugated polymers.

These additional results therefore extend the scope of our mechanism beyond individual materials and support its general applicability to conjugated polymer-based electrochemical systems.

We have added related discussions on page 15 (highlighted texts).

Figure S29. a, Schematic illustration of the electropolymerization of EDOT and the redox mechanism of PEDOT, where A^- represents the counter anions from the electrolyte that compensate the positive charges on the oxidized PEDOT backbone. CV curves of PEDOT in **b**, 2 M ZnSO₄ and **c**, 2 M Zn(OTf)₂, recorded at scan rates from 0.01 to 0.1 V s⁻¹.

Reviewer 2

In this work, the authors developed a sub-millimeter monolithic device that unifies energy storage and actuation in a single, synergistic platform. Through hydration control, the performance of both micro-battery and actuator has been improved, providing a representative example using multifunctional materials to construct integrated devices. Therefore, it could be recommended for publication in Nature Communications. Before, some issues still need to be addressed.

1. The long-term cycle performance should be evaluated for the single micro-battery. Further, the reason for rapid capacity degradation of dual-cell micro-battery in initial cycles (Figure 4d) should be explained. In addition, at the current of 5 μ A, why is the capacity of dual-cell micro-battery in Figure 4b and 4d different?

Reply: Thank you for your valuable comment. We focused on the dual-cell configuration because it provides a higher capacity than the single-cell configuration and is therefore more relevant for practical operation. Moreover, the long-term cycling stability of the dual-cell device is more critical, as the self-folding process introduces mechanical stress in the current collector, particularly at the hinge region where a small curvature is maintained during prolonged cycling. Any mechanical or electrical failure of the current collector at this location would directly lead to device failure. For this reason, we chose to present the cycling performance of the dual-cell configuration. Nevertheless, we have also investigated the stability of PANI in a single-cell configuration. As shown in Figure S20, the device maintains a capacity retention of 70% over 100 cycles, comparable to the dual-cell configuration.

The initial capacity degradation observed in both the single- and dual-cell configurations can be primarily attributed to the self-folding process carried out in 0.1 M NaOH. We examined the impedance of PANI before and after 10 min immersion in NaOH (corresponding to the self-folding duration) and found a clear increase in resistance (Figure S3a), which is attributed to partial deprotonation of PANI. Meanwhile, the increase in phase angle indicates that interfacial parasitic processes become non-negligible. Accordingly, the early capacity fade mainly originates from an initial re-equilibration process, including re-doping and ion exchange, of partially deprotonated PANI after NaOH exposure, as further evidenced by the relatively low Coulombic efficiency (~95%) in the initial cycles (Figure S21). The hydroxide-induced re-equilibration is further reflected in the dQ/dV plot (Figure S22), in which a strong reduction peak appears in the first cycle, corresponding to irreversible hydroxide extraction, and fades

with continued cycling. We have recognized this issue and, in the following electrochemical evaluations, report the stabilized capacity after this re-equilibration stage, thereby avoiding overestimation of the effective capacity.

This early-stage behaviour may lead to confusion when comparing long-term cycling data with rate performance. During the rate capability test, the discharge capacity at 5 μA is 0.37 μAh (Figure 4b). Under the same current in the cycling experiment (Figure 4d), the capacity recorded after 30 cycles falls in the range of 0.33 to 0.36 μAh (also shown in Figure S21), confirming that the stabilized capacity in long-term cycling is consistent with the rate performance measured at the same current.

We have added related discussions on pages 5-6 and 12 (highlighted texts).

Figure S20. Capacity retention of one single cell cycled at 2.5 μA .

Figure S21. Stability of the dual-cell microbattery in the initial 50 cycles at 5 μA .

Figure S22. Differential capacity (dQ/dV)– V profiles of the dual-cell microbattery extracted from galvanostatic charge–discharge measurement at $5 \mu\text{A}$ (Figure S21).

2. The areal capacity of single cell should be provided, not just a lifetime capacity.

Reply: Thanks for your suggestion. We have provided the areal (footprint) capacity of the microbattery in the main text. For the rate performance, at a discharge current of $1 \mu\text{A}$, the microbattery delivers an actual capacity of $0.59 \mu\text{Ah}$ ($1.05 \mu\text{Ah mm}^{-2}$). Even at a high current of $20 \mu\text{A}$, the device still retains a capacity of $0.19 \mu\text{Ah}$ ($0.34 \mu\text{Ah mm}^{-2}$). Notably, the discharge plateau remains visible at around 1.14 V , indicating that redox activity is still sustained at higher rates. Upon reducing the current back to $1 \mu\text{A}$, the capacity recovers to $0.53 \mu\text{Ah}$ ($0.95 \mu\text{Ah mm}^{-2}$), demonstrating excellent reversibility with a retention of 90% compared to the initial cycle. Long-term cycling at $5 \mu\text{A}$ further confirms the device's stability: over 2200 cycles are achieved with a capacity of $0.29 \mu\text{Ah}$ ($0.52 \mu\text{Ah mm}^{-2}$) and nearly 100% Coulombic efficiency after the initial conditioning period (Figure 4d).

We have added related values on pages 11-12 (highlighted texts).

3. In a very small space, the precise addition and confinement is usually a challenge. The authors should introduce how to realize this in detail.

Reply: Thank you for acknowledging this important challenge. We fully agree that, within a sub-millimeter footprint, the precise addition and confinement of electrode and electrolyte materials are technically demanding. We would like to emphasize, however, that a central motivation of this work is to develop an electrochemical device that unifies energy storage and actuation in an open system, rather than in a sealed, closed configuration. If strict isolation and encapsulation were required, the unifying principle would become less meaningful, as the energy storage unit could always be separated as an independent microbattery. In contrast, our

concept is intentionally designed to operate in open environments, where the surrounding solution can directly participate in electrochemical processes and mechanical response.

Accordingly, our device is aimed at applications where actuation in open media is essential, as exemplified in Figure 5a, where the actuator regulates fluidic flow. In this context, we further extended the electrolyte-controlled hydration mechanism to a more general aqueous solution (Figures S30 and S31) rather than to a well-defined electrolyte. The device maintains stable and reproducible operation in the general solution, demonstrating that the hydration-regulated mechanism is not specific to one particular electrolyte formulation, but reflects a more general principle.

We also appreciate the reviewer's point regarding electrolyte control during electrochemical evaluation, as variations in electrolyte volume can indeed influence device performance. In our experiments, although the device is designed to function in open environments, we used a PDMS mold (Figure S19) to define and control the electrolyte volume (approximately 5 μL) during characterization. This ensures good experimental reproducibility while preserving the open-system nature of the device.

We have added related discussions on page 11 (highlighted texts).

Figure S30. Optical images of PPy actuation in **a**, 0.9 wt% NaCl and **b**, DPBS. Scale bar: 250 μm .

Figure S31. **a**, Chemical structure of ACES (N-(2-acetamido)-2-aminoethanesulfonic acid). CV curves of PANI in **b**, 0.9 wt% NaCl, **c**, 0.9 wt% NaCl + 200 mM ACES, **d**, DPBS, and **e**, DPBS + 200 mM ACES, recorded at a scan rate of 0.8 mV s^{-1} .

Figure S19. Optical images of the PDMS encapsulation for microbattery performance tests. Scale bars: $500 \mu\text{m}$.

4. The linear sweep voltammetry should be provided for the two electrolytes, to support whether the water decomposition happen at higher voltage.

Reply: Thanks for your suggestion. To clarify the electrochemical stability of the two aqueous electrolytes, we performed linear sweep voltammetry measurements for 2 M ZnSO₄ and 2 M Zn(OTf)₂ (Figure S11). The experiments were conducted at a scan rate of 5 mV s⁻¹, using a Pt mesh counter electrode, stainless-steel working electrode, and an Ag/AgCl reference electrode. Both electrolytes are stable between -0.8 and 1.35 V. Therefore, the degradation of PANI cannot be attributed to electrolyte decomposition, but instead arises from hydration water incorporated into the polymer backbone, as evidenced by the Raman spectra (Figure 2c).

We have added related discussions on page 7 (highlighted texts).

Figure S11. LSV profiles of 2 M ZnSO₄ and 2 M Zn(OTf)₂ electrolytes recorded at a scan rate of 5 mV s⁻¹.

5. The PANI cathode shows obvious performance difference in two electrolytes. How is the Zn anode? Does the Zn anode give different reversibility in the two electrolytes? Would Zn anode influence the cycle of micro-battery?

Reply: Thank you for your insightful comments. We fully agree that the Zn anode can influence the overall electrochemical performance, and indeed Zn(OTf)₂ has often been reported to provide better Zn stability than ZnSO₄.^{1,2} To clarify the contribution of the Zn anode in our system, we have added Zn||Zn symmetric cell cycling tests in both 2 M ZnSO₄ and 2 M Zn(OTf)₂ electrolytes. As shown in Figure S9, both electrolytes exhibit comparable cycling stability, although ZnSO₄ shows a slightly higher overpotential. The similar long-term stability of the symmetric cells indicates that the difference observed in the full devices cannot be primarily attributed to the Zn anode behavior. We therefore conclude that the performance

divergence in the full cells mainly originates from the PANI electrode and its electrolyte-dependent interfacial chemistry, rather than from instability of the Zn metal itself.

We have added related discussions on page 7 (highlighted texts).

Figure S9. Comparison of galvanostatic cycling of Zn||Zn symmetric cells in 2 M ZnSO₄ and 2 M Zn(OTf)₂ electrolytes at 1 mA cm⁻² with 1 mAh cm⁻².

Reviewer 3

In this paper, Zhang et al. developed a sub-millimeter monolithic device that unifies energy storage and simple actuation functions. Importantly, the authors innovatively used triflate OTf- to mitigate the hydration issue of the conjugated polymer polyaniline (PANI), thereby improving its cycling stability and device lifetime. A thorough analysis based on in-operando Raman spectroscopy, time-resolved mass spectrometry, and various simulation approaches has been conducted to elucidate the mechanism and effects of water in conjugated polymers. Further, the developed dual-cell and multiple-cell microbatteries have shown great potential in the important areas of miniature batteries and microrobots. Therefore, I would recommend the publication of this work.

Before acceptance, there are some major issues that need to be addressed.

Major Issues:

1. Figure 1 should be significantly revised. Here are my concerns and suggestions. The main innovation of addressing PANI hydration and degradation is not shown. Fig. 1a vaguely explains the mechanism of the biological muscle fascicle and is not directly connected to the device structure. I suggest moving it to the SI. The battery reactions and actuator mechanisms are not shown, which could be unclear to non-specialist readers. The device's internal structure after folding is unclear to me. How the battery is connected to the actuators is unclear.

Reply: Thank you for pointing out these issues, which may indeed cause confusion. Figure 1a is intended as a conceptual motivation for our work, illustrating the advantage of using microfabrication to create a multifunctional electrochemical microdevice. In biological systems, energy supply and function are often intrinsically integrated at the local level, rather than being separated into centralized power sources and individual functional units, as is typical in modern electronic systems. While the functional density of today's microsystems is already very high, the corresponding microscale energy supply still lags far behind, creating a fundamental bottleneck.

Here, we propose a strategy of localized integration of energy storage and function, enabling distributed components that each carry both energy and functionality. This approach can fundamentally relax the constraints of microscale energy storage, which is often insufficient to support fully integrated systems when centralized. Within this framework, we pursue two complementary directions: (i) increasing local energy density by stacking multiple energy storage units, and (ii) reducing functional power requirements, exemplified here by ultralow-

power electrochemical actuation. Importantly, we identify a unified mechanism (electrolyte-controlled hydration) that simultaneously enhances the stability of energy storage and reduces the power consumption of actuation in conjugated polymer systems.

Nevertheless, we agree with the reviewer that the original presentation did not clearly convey the underlying mechanisms of energy storage and actuation, which could present a barrier for readers. To address this, we have added a general schematic introducing charge storage in conjugated polymers in Figure 1b, together with illustration of the device structure after folding. The connection between the energy-storage unit and the actuator follows the same scheme as the interconnection between the two sub-cells in the dual-cell configuration (Figure 1f). Individual leads are connected to each actuator and routed to contact pads at the edge of the substrate. The on/off operation is controlled by an external switch (Figure S1).

We have revised Figure 1 and added related discussions on pages 3-5 (highlighted texts).

Figure S1. Schematic illustration of the circuit configuration of the dual-cell Zn-PANI microbattery connected to PPy actuators.

2. It is also unclear to me why the authors focused on a dual-cell microbattery design, when they could hugely increase the cell numbers, such as a five-cell microbattery shown in Fig. S3. Other structural dimensions are also vaguely explained, e.g., 820 μm long and 166 μm wide. I wonder what the influence of battery size is. What is the minimal size the authors can achieve, while keeping the rationale of the PANI device valid?

Reply: Thank you for pointing out that a five-cell configuration is in principle achievable and for raising this important question. We chose to focus on the dual-cell configuration primarily

based on the trade-off between architectural complexity and microfabrication yield. Because this work deals with microscale devices, fabrication yield is a critical but often overlooked factor. Although five-cell structures can indeed be fabricated, the yield is currently very low. Even minor misalignment during the self-folding process can lead to device failure, not to mention the additional challenges associated with maintaining electrochemical reproducibility during materials synthesis and cycling. For example, overgrowth of materials at some electrodes often occurs (Figure S6). In contrast, the dual-cell configuration can be fabricated with a high yield ($> 90\%$), while still capturing the essential architectural and functional features of the system. Therefore, although the five-cell configuration demonstrates promising scalability, it would require substantial further advances in both microengineering precision and materials process control, which are beyond the scope of the present study that focuses on unifying electrochemical energy storage and actuation within one device.

We also appreciate the reviewer's question regarding the impact of battery size on performance. In practice, depositing thick, uniform, and electrochemically reproducible active layers becomes increasingly difficult as the footprint shrinks, which necessitates a reduction in electrode thickness at smaller dimensions. We have therefore added a quantitative comparison of footprint-normalized capacity as a function of electrode size. As shown in Figure S7, the footprint capacity decreases as the footprint is reduced. The decline is not fully linear, but becomes much more pronounced below $300\ \mu\text{m}$. This result highlights that, at very small dimensions, the absolute capacity drops sharply and quickly becomes impractical, further underscoring the importance of architectural design and functional integration at the microscale. We have now labeled the characteristic dimensions in the relevant figures for clarity. We have added related discussions on page 6 (highlighted texts).

Figure S6. Optical images of electrodeposited Zn and PANI on the five-cell design. Scale bars: $500\ \mu\text{m}$.

Figure S7. Capacity as a function of the footprint area of a single electrode for Zn-PANI batteries with lateral dimensions ranging from $2 \times 2 \text{ mm}^2$ down to $0.075 \times 0.075 \text{ mm}^2$. Optical microscopy images show Zn-PANI batteries corresponding to the six different electrode sizes used in the capacity comparison.

3. By replacing sulphate anions with triflate, PANI shows a significant improvement in its stability. What is the aqueous environment for this test? What is the device encapsulation? How will the device perform in physiological environments?

Reply: Thank you for comments. The aqueous solution is 2 M $\text{Zn}(\text{OTf})_2$ and 2 M ZnSO_4 . These two electrolytes were intentionally selected as model systems to compare the effect of a hydrophobic weakly coordinating anion (OTf^-) and a strongly hydrated anion (SO_4^{2-}) on the electrochemical stability of conjugated polymers.

A central motivation of this work is to develop an electrochemical device that unifies energy storage and actuation in an open system, rather than in a sealed, closed configuration. If strict isolation and encapsulation were required, the unifying principle would become less meaningful, as the energy storage unit could always be separated as an independent microbattery. In contrast, our concept is intentionally designed to operate in open environments, where the surrounding solution can directly participate in electrochemical processes and mechanical response. Accordingly, our device is aimed at applications where actuation in open media is essential, as exemplified in Figure 5a, where the actuator regulates fluidic flow. We have considered electrolyte control during electrochemical evaluation, as variations in

electrolyte volume can indeed influence device performance. In our experiments, although the device is designed to function in open environments, we used a PDMS mold (Figure S19) to define and control the electrolyte volume (approximately 5 μL) during characterization. This ensures good experimental reproducibility while preserving the open-system nature of the device.

Regarding physiological environments, it has been widely reported that PPy actuators can operate in NaCl and PBS solutions.³⁻⁶ In this work, we also confirmed that PPy actuators remain functional in physiological saline (0.9 wt% NaCl) and DPBS, as shown in Figure S30. Batteries, however, are fundamentally more demanding. Physiological saline and DPBS are not suitable battery electrolytes because they contain extremely low concentrations of electrochemically active ions and lack the solvation structures required to support reversible, high-capacity redox reactions. To nevertheless probe whether the hydration-controlled mechanism itself is transferable to biocompatible chemical environments, we introduced a biocompatible sulfonate-based molecule, N-(2-acetamido)-2-aminoethanesulfonic acid (ACES),⁷ as a weakly hydrated additive. As shown in Figure S31, PANI exhibits poor reversibility in 0.9 wt% NaCl and DPBS due to severe water-driven deprotonation and degradation. After adding ACES, the electrochemical reversibility of PANI is partially restored in both media. This demonstrates that even in physiologically relevant solutions, introducing weakly coordinating sulfonate species can actively reshape the local solvation and hydration environment, suppress water-induced degradation, and recover reversible redox behavior. Importantly, this result shows that our mechanism is not specific to OTf, nor to Zn-based electrolytes, but reflects a general and chemically extensible principle: hydration can be deliberately engineered through weakly coordinating, hydration-modulating species to control the electrochemistry of conjugated polymers.

These experiments therefore extend the scope of our concept from a specific electrolyte system to a broader design framework, in which interfacial hydration can be tuned using different anions or zwitterionic/biocompatible molecules to regulate stability, kinetics, and reversibility across diverse environments. It should be noted that without a sufficient concentration of Zn^{2+} , battery performance cannot be guaranteed; however, this does not invalidate the hydration-control principle disclosed here. Rather, it highlights a critical design consideration (local solvation and hydration environments) that can effectively enhance electrochemical stability and reversibility. In practical applications, translating this principle into physiologically compatible systems will require substantial further materials innovation, which is beyond the

scope of this work that focuses on revealing the fundamental interplay between conjugated polymers and electrolyte-controlled hydration.

We have added related discussions on pages 11 and 16 (highlighted texts).

Figure S19. Optical images of the PDMS encapsulation for microbattery performance tests.

Scale bars: 500 μm .

Figure S30. Optical images of PPy actuation in **a**, 0.9 wt% NaCl and **b**, DPBS. Scale bar: 250 μm .

Figure S31. a, Chemical structure of ACES (N-(2-acetamido)-2-aminoethanesulfonic acid). CV curves of PANI in **b**, 0.9 wt% NaCl, **c**, 0.9 wt% NaCl + 200 mM ACES, **d**, DPBS, and **e**, DPBS + 200 mM ACES, recorded at a scan rate of 0.8 mV s^{-1} .

4. The shifts of Raman peaks in Fig. 3b seem marginal. How valid is the methodology to differentiate SSIP and CIP states?

Reply: Thank you for this comment. To avoid any potential misunderstanding, we have clarified in the main text that our discussion of SSIP and CIP is intended to be qualitative rather than quantitative. The Raman measurements in Figure 3b were performed operando, that is, on the same electrode and at the same location while the potential was continuously varied. Therefore, the observed peak shifts and shape evolution arise from changes in the local chemical environment at the polymer–electrolyte interface, rather than from variations between different samples or different measurement spots.

In Figure 3b, the SO_4^{2-} band ($\sim 976\text{--}988 \text{ cm}^{-1}$) recorded on PANI in ZnSO_4 shows a clear peak-shape evolution with increasing potential. The high-frequency shoulder ($\sim 984 \text{ cm}^{-1}$) is assigned to CIP, whereas the low-frequency component ($\sim 980 \text{ cm}^{-1}$) corresponds to SSIP. This peak-

shape evolution is consistent with a transition from solvent-separated ion pairs (SSIP) to contact ion pairs (CIP) of SO_4^{2-} , as reported in previous ZnSO_4 Raman studies.⁸⁻¹⁰ By contrast, the OTf band (bottom panel) exhibits only a marginal shift, indicating a more stable interaction with the PANI backbone and no distinct evolution between SSIP and CIP states.

We have added related discussions on page 10 (highlighted texts).

5. The final application in driving actuation is interesting. However, I cannot see a strong relevance to muscle fascicles (see my comment 1). The one-off actuation and the use of a dual-cell design, instead of others, should be further explained.

Reply: Thanks for the insightful comments. Muscle fascicles are an example of biological systems where energy supply and function are often intrinsically integrated at the local level, rather than being separated into centralized power sources and individual functional units, as is typical in modern electronic systems. While the functional density of today's microsystems is already very high, the corresponding microscale energy supply still lags far behind, creating a fundamental bottleneck. The concept of our device was inspired by muscle fascicles, where localized electrical stimulation triggers localized contraction. In analogy, our system aims to integrate local energy supply with local actuation in a single microscale unit. We have updated Figure 1 for improved clarity.

The actuation powered by the microbattery is not one-off. It is reversible and repeatable, as demonstrated by the cyclic actuation shown in Supplementary Video S2.

We chose to focus on the dual-cell configuration primarily based on the trade-off between architectural complexity and microfabrication yield. Because this work deals with microscale devices, fabrication yield is a critical but often overlooked factor. Although five-cell structures can indeed be fabricated, the yield is currently very low. Even minor misalignment during the self-folding process can lead to device failure, not to mention the additional challenges associated with maintaining electrochemical reproducibility during materials synthesis and cycling. In contrast, the dual-cell configuration can be fabricated with a much higher yield, while still capturing the essential architectural and functional features of the system.

We have added related discussions on page 3 (highlighted texts).

6. Recent work has shown the importance of solid-state and soft microbatteries in driving ionic movements and various biological modulation, as well as potential applications in microrobots.

The authors are recommended to discuss more about the potential applications of their microbatteries by comparison with other work.

Reply: Thank you for this valuable suggestion. As illustrated in Figure 1a, our motivation is to develop a localized integration of energy storage and function, inspired by biological systems that often couple energy supply and functionality at the local level. Pioneering studies have demonstrated microscale lithium-based power sources for neuromodulation¹¹ and bioresorbable pacemakers¹², where batteries are successfully used either to directly deliver localized electrical stimulation or as onboard power units to enable self-powered therapeutic devices. We highly appreciate these outstanding works, which clearly establish the feasibility and biomedical relevance of microscale batteries; complementary to these function-oriented advances, our work offers a different perspective by focusing on the electrochemical and materials chemistry foundations, revealing how electrolyte–polymer interactions and hydration control can be leveraged to systematically improve stability and performance.

We have added related discussions on pages 15-16 (highlighted texts).

Reviewer 4

This manuscript reports a monolithic, sub-millimeter integrated device that combines conjugated-polymer-based energy storage with electrochemically driven actuation (PPy micro-actuators). Using a broad experimental toolkit, the authors propose a design principle: disrupting ion hydration shells reduces water ingress into the conjugated polymer, suppresses PANI hydrolysis, and thereby improves storage stability as well as actuation bandwidth/energy efficiency. Linking solvation-structure engineering to both durability and performance at the sub-millimeter scale is timely and impactful for microsystems that co-localize power and motion (e.g., microrobotics, lab-on-chip). The concept is compelling, experimentally rich, and cross-disciplinary—appropriate for Nature Communications. Overall, this is one of the most beautiful studies I have reviewed in this area, as I know it is always very challenging to integrate battery systems into such small scales while still maintaining decent performance. That said, several issues still should be considered before publication.

1. The manuscript demonstrates application-relevant functions (e.g., LED lighting, powering low-power digital watches), but many of these exciting results are in the Supplementary Information. To improve readability and impact, the authors are suggested to migrate a subset into the main figures—for example, consider integrating current Fig. S2 into Fig. 1f.

Reply: Thanks for the suggestion. To improve the readability and clarity of Figure 1, we have significantly revised it. First, we clarified that the muscle fascicle serves as the conceptual motivation for creating a system in which local actuation and energy storage are unified through electrochemical processes, thereby avoiding long-distance energy transfer through wires from separated, centralized power sources, as is common in most modern microscale system designs. The foundation of this unified electrochemical approach lies in the use of conjugated polymers, which are inherently multifunctional. As suggested, we also added quoted Figure S2 to the main figure set to illustrate the parallel fabrication process of the self-folding device.

2. The folding process is novel and crucial for efficient space utilization, but it remains difficult for me to follow without watching the video. The schematics are likely due to an unhelpful viewing angle. The authors are suggested to add a clearer schematic (e.g., an alternate perspective or step-wise panels with numbered stages and motion arrows) that makes the self-folding mechanism and hinge locations unambiguous.

Reply: To improve the clarity of the self-folding process, we added more explanatory labels to Figure 1f, illustrating three key stages of the folding sequence with representative snapshots. Panel f.1 shows the planar precursor design with the folding hinges highlighted. Panel f.2 illustrates that, after removal of the sacrificial layer, the planar thin-film stack becomes freestanding and the central hinge is triggered first. This initial folding step forms the shared layer in the stacked dual-cell configuration. Subsequently, both side hinges are activated, ultimately folding the structure into two enclosed boxes that serve as the individual battery units.

We have added the related discussion on page 5 (highlighted texts).

3. Please add scale bars (and, if feasible, frame-overlay timestamps) to all Supplementary Videos and main figures.

Reply: Thanks for your suggestion. We have added scale bars to all main figures, Supplementary Figures and Videos. We have also included frame-overlay timestamps to all Supplementary Videos.

4. The red line in Fig. 4a requires a precise definition in the caption and/or main text (what quantity it represents, how it is derived, and its role in the analysis).

Reply: Thanks for this valuable suggestion. We have revised Figure 4a and its caption to define the red lines in the main text. Specifically, the red lines denote the folding hinges of the dual-cell microbattery. After self-folding, the planar pattern transforms into the top and bottom cells, thereby defining the dual-cell configuration.

References

1. Zhang, N. *et al.* Cation-deficient spinel ZnMn_2O_4 cathode in $\text{Zn}(\text{CF}_3\text{SO}_3)_2$ electrolyte for rechargeable aqueous Zn-ion battery. *J. Am. Chem. Soc.* **138**, 12894–12901 (2016).
2. Yuan, D. *et al.* Anion texturing towards dendrite-free Zn anode for aqueous rechargeable batteries. *Angew. Chem. Int. Ed.* **60**, 7213–7219 (2021).
3. Martinez, J. G., Otero, T. F. & Jager, E. W. H. Effect of the electrolyte concentration and substrate on conducting polymer actuators. *Langmuir* **30**, 3894–3904 (2014).
4. Cao, D., Martinez, J. G., Satoshi Hara, E. & Jager, E. W. H. Soft actuators that self-create bone for biohybrid (micro)robotics. in *2022 International Conference on Manipulation, Automation and Robotics at Small Scales (MARSS)* 1–6 (2022).
5. Beregoi, M., Evangelidis, A., Diculescu, V. C., Iovu, H. & Enculescu, I. Polypyrrole actuator based on electrospun microribbons. *ACS Appl. Mater. Interfaces* **9**, 38068–38075 (2017).
6. Arnaboldi, S. *et al.* Bi-enzymatic chemo-mechanical feedback loop for continuous self-sustained actuation of conducting polymers. *Nat. Commun.* **14**, 6390 (2023).
7. Singh, G. *et al.* Good's buffer based highly biocompatible ionic liquid modified PLGA nanoparticles for the selective uptake in cancer cells. *Mater. Chem. Front.* **7**, 6213–6228 (2023).
8. Yang, H. *et al.* A metal–organic framework as a multifunctional ionic sieve membrane for long-life aqueous zinc–iodide batteries. *Adv. Mater.* **32**, 2004240 (2020).
9. Gou, Q. *et al.* Electrolyte regulation of bio-inspired zincophilic additive toward high-performance dendrite-free aqueous zinc-ion batteries. *Small* **19**, 2207502 (2023).
10. Miao, Z. *et al.* Tailoring local electrolyte solvation structure via a mesoporous molecular sieve for dendrite-free zinc batteries. *Adv. Funct. Mater.* **32**, 2111635 (2022).

11. Zhang, Y. *et al.* A microscale soft lithium-ion battery for tissue stimulation. *Nat. Chem. Eng.* **1**, 691–701 (2024).
12. Zhang, Y. *et al.* Millimetre-scale bioresorbable optoelectronic systems for electrotherapy. *Nature* **640**, 77–86 (2025).

Reply to Reviewers

Reviewer 3

My only remaining concern is that the use of external control/logic circuits to reversibly charge-discharge micro-actuators in Fig 5 should be clearly mentioned in the main text. I would be happy to recommend publication after this revision.

Reply: Thanks for the comment. We have further clarified the control of actuators with a circuit diagram in Supplementary Fig. 1. Specifically, when switches S2 and S3 are closed, the actuator is injected with ions, inducing bending. Subsequently, disconnecting S2 and S3, while closing switches S1 and S4, reverses the polarity of the actuator, leading to the release of ions and consequently relaxation. Related discussions have been added to the main text on pages 10 and 15.